# ANN-Based LiDAR Positioning System for B5G

**DOI:** 10.3390/mi15050620

**Published:** 2024-05-04

**Authors:** Egidio Raimundo Neto, Matheus Ferreira Silva, Tomás P. V. Andrade, Arismar Cerqueira Sodré Junior

**Affiliations:** Laboratory WOCA, National Institute of Telecommunications (Inatel), 510 João de Camargo Av., Santa Rita do Sapucaí 37540-000, MG, Brazil; matheus.ferreira@get.inatel.br (M.F.S.); tomasvillena@dtel.inatel.br (T.P.V.A.); arismar@inatel.br (A.C.S.J.)

**Keywords:** 2D-LiDAR, B5G, artificial neural network, 6G, positioning, sensing

## Abstract

This work reports the development of an efficient and precise indoor positioning system utilizing two-dimensional (2D) light detection and ranging (LiDAR) technology, aiming to address the challenging sensing and positioning requirements of the beyond fifth-generation (B5G) mobile networks. The core of this work is the implementation of a 2D-LiDAR system enhanced by an artificial neural network (ANN), chosen due to its robustness against electromagnetic interference and higher accuracy over traditional radiofrequency signal-based methods. The proposed system uses 2D-LiDAR sensors for data acquisition and digital filters for signal improvement. Moreover, a camera and an image-processing algorithm are used to automate the labeling of samples that will be used to train the ANN by means of indicating the regions where the pedestrians are positioned. This accurate positioning information is essential for the optimization of B5G network operation, including the control of antenna arrays and reconfigurable intelligent surfaces (RIS). The experimental validation demonstrates the efficiency of mapping pedestrian locations with a precision of up to 98.787%, accuracy of 95.25%, recall of 98.537%, and an F1 score of 98.571%. These results show that the proposed system has the potential to solve the problem of sensing and positioning in indoor environments with high reliability and accuracy.

## 1. Introduction

The genesis of the mobile telecommunications industry can be traced back to the advent of the first generation of mobile communications systems (1G), characterized by analog cellular systems. Notably, the Advanced Mobile Phone System (AMPS) in the United States and the Nordic Mobile Telephone (NMT) in Europe epitomized this era, initially facilitating mobile voice calls circa 1980. Subsequently, the industry has witnessed the introduction of a new generation of mobile communications approximately every decade. The AMPS systems were superseded by the second generation of mobile communications systems (2G), which heralded the era of second-generation digital cellular networks around 1990. Amidst various competing systems, the Global System for Mobiles (GSM) [1] emerged as a commercial triumph, enabling over a billion individuals to access mobile voice, short text messages, and low-rate data services.

The third generation of mobile communications Systems (3G), leveraging a groundbreaking technology known as code-division multiple access (CDMA), encompassed wide-band code-division multiple access (WCDMA), CDMA2000, and time division-synchronous code division multiple access (TD-SCDMA). These systems were introduced in 2001, marking the inception of high-speed data access capabilities at rates of several megabits per second [2]. The commercial deployment of long-term evolution (LTE) networks in December 2009 in Stockholm and Oslo signified the advent of the fourth generation of mobile communications systems (4G), offering the world’s inaugural mobile broadband service. The 4G architecture, underpinned by the synergistic integration of multiple-input multiple-output (MIMO) and orthogonal frequency division multiplexing (OFDM), catalyzed the smartphone revolution and the exponential growth of the mobile internet sector, which now commands a market moving trillions of dollars annually [3].

Diverging from its predecessors, which primarily aimed at enhancing network capabilities, the fifth generation of mobile communication systems (5G) broadens the spectrum of mobile communication services to encompass not only human interactions but also connectivity between humans, machines, and things, extending its reach to vertical industries. This expansion significantly increases the potential scale of mobile subscriptions, from billions of global users to virtually innumerable interconnections. Moreover, 5G introduces enhanced mobile broadband (eMBB), achieving peak data rates up to 10 Gbps, ultra-reliable low-latency communication (uRLLC) with delays reduced to as low as 1 ms, and massive machine type communication (mMTC), supporting connectivity for over 100 times more devices per unit area compared to 4G. The anticipated network reliability and availability exceed 99.999% [4]. A key innovation of 5G, network softwarization, facilitates network dynamism, programmability, and abstraction [5]. The advent of 5G has unlocked new applications across diverse domains, including virtual reality (VR), augmented reality (AR), mixed reality (MR), the Internet of Things (IoT), Industry 4.0, and autonomous vehicles [6,7,8].

While the global deployment of 5G is actively ongoing, both the academic and industrial sectors are progressively shifting their focus towards the horizon beyond 5G, commonly referred to as B5G or the sixth generation of mobile communications systems (6G). This forward-looking perspective is driven by the ambition to fulfill the burgeoning demands of information and communication technology (ICT) by the year 2030. In alignment with this vision, a number of seminal initiatives concerning 6G networks have already commenced [9]. In July 2018, a focus group named Technologies for the 2030 Network was inaugurated within the standardization sector of the International Telecommunication Union (ITU-T). This group is dedicated to exploring the potential capabilities of future networks for the year 2030 and beyond. Such networks are anticipated to underpin a plethora of innovative and visionary applications, including but not limited to holographic communications, ubiquitous intelligence, the tactile internet, multisensory experiences, and the digital twin concept [10]. Further emphasizing the commitment to this futuristic vision, the radiocommunication sector of the International Telecommunication Union (ITU-R) convened in February 2020 and resolved to initiate a comprehensive study on the forthcoming technological trends. This study aims to guide the future evolution of international mobile telecommunications (IMT) [11].

Recent advancements in the field of communications have heralded the introduction of several innovative concepts, including edge intelligence (EI), the expansion of communication frequencies beyond sub-6GHz to the terahertz (THz) range, non-orthogonal multiple access (NOMA), and large intelligent surfaces (LIS) [12,13]. These emerging concepts are on the cusp of evolving into fully fledged technologies poised to catalyze the next wave of advancements in communication networks. Concurrently, a spectrum of applications, such as holographic telepresence (HT), unmanned aerial vehicles (UAV), extended reality (XR), Smart Grid 2.0, Industry 5.0, and ventures into space and deep-sea tourism, are anticipated to dominate as the principal applications within future communication frameworks. Nonetheless, the stringent requirements posed by these applications—including ultra-high data rates, real-time access to substantial computational resources, exceedingly low latency, precise location tracking and detection, alongside unparalleled reliability and availability—surpass the network capabilities currently envisaged by 5G [14,15]. Moreover, the IoT, which finds its enablement through 5G technology, is evolving into the Internet of Everything (IoE). This progression aims to interconnect an extensive array of sensors, devices, and cyber-physical systems (CPS), stretching beyond the confines of 5G’s capabilities. Such a paradigm shift has galvanized the research community towards the conceptualization of 6G mobile communication networks. 6G is poised to harness the potential of new communication technologies, fully support the emergence of novel applications, facilitate connectivity among an unprecedented number of devices, and ensure real-time access to potent computational and storage resources.

The expanding demand for location-based services (LBS) has underscored the need for accurate real-time positioning, especially in indoor environments where the global positioning system (GPS) is inadequate. Indoor environments present unique challenges due to their complex and varied nature, often characterized by the presence or absence of stationary and moving objects, such as furniture and people. These complexities significantly affect propagation under both line-of-sight (LOS) and non-line-of-sight (NLOS) conditions, causing unpredictable alterations such as attenuation, scattering, shadowing, and the creation of blind spots. These factors collectively degrade the accuracy of indoor positioning [16].

The intense growth of indoor positioning systems (IPS) [17] has been driven by techniques based on different metrics, including the received signal strength indicator (RSSI) [18]; time-of-arrival (ToA) [19]; time difference of arrival (TDoA) [20]; and angle of arrival (AoA) [21]. Furthermore, channel state information (CSI) techniques [22] have been proposed utilizing wireless access technologies, such as Wi-Fi [23], Bluetooth [24], ultra-wideband (UWB) [25] and radio-frequency identification (RFID) [26]. These methods, while innovative, face several challenges such as poor accuracy, high computational complexity, and unreliability, compounded by the limitations in processing power of most positioning devices [27]. Global navigation satellite systems (GNSS), commonly used in outdoor environments for robot localization, suffer from limited availability in dense urban and rural environments. Similarly, radio-based technologies such as Bluetooth, UWB, Wi-Fi, and 4G LTE have been used for positioning purposes with varying degrees of success. However, the quality and availability of fix measurements in GNSS and other radio-based technologies may become obsolete due to environmental factors, such as obstructions and multi-path effects, particularly in urban settings [28].

Recent years have witnessed a significant success of artificial intelligence (AI) and machine learning (ML) algorithms in indoor localization. The core advantage of these AI/ML approaches lies in their capability to effectively make decisions using observed data without the need for precise mathematical formulations. This has been particularly effective in addressing the aforementioned challenges in IPS [29]. ML has also been instrumental in fusing multi-dimensional data collected from multiple positioning sensors, technologies, and methods. Both supervised and unsupervised learning techniques have been applied for generating fusion weights [30]. Notably, unsupervised ML fusion techniques are superior as they compute weights in real time without the need for offline training [31]. Transfer learning, another aspect of ML, plays a significant role in enhancing the scalability of fingerprint-based localization systems, especially in dynamically changing indoor environments. It allows for quick learning in new environments by leveraging previously acquired knowledge, thus enabling more efficient and scalable IPS models [32].

The evolution from the 5G to the 6G is marked by the integration of communication and sensing into a unified framework, known as joint communication and sensing (JCAS). This integration is essential for creating an authentic digital representation of the physical world, effectively turning the network into a sensor that augments human intelligence. The introduction of wider bandwidth systems in 5G, combined with dense small cell deployments, has laid the groundwork for using these systems for advanced sensing applications [33]. Simultaneous localization and mapping (SLAM) is a fundamental research topic in robotics, significantly influencing the prolonged operation of mobile robots [34]. Traditional SLAM methods have utilized LiDAR alongside visual-based methods to build a representative map while maintaining the robot localized within that map. These approaches have gained prominence due to their ability to create high-quality maps of environments combined with accurate pose estimators [35]. LiDAR-based methods, along with inertial and visual methods, offer significant advantages. They are less prone to drift and can adapt to environmental changes and varying illumination conditions. LiDAR sensors are effective in creating detailed environmental maps by obtaining physical ranges between the robot and its surroundings or detecting suitable features to track within the target area. This capability is crucial, especially in outdoor areas where uncertainty increases due to uneven terrain, dense urban structures, and temporary environmental changes such as fog, rain, and dust [36].

Recent advancements in LiDAR technology have catalyzed transformative changes across numerous sectors, enhancing both reliability and precision for a myriad of applications, including but not limited to autonomous driving and human–robot interactions [37,38]. The acquisition of three-dimensional (3D) data through LiDAR furnishes a detailed 360-degree environmental representation, empowering robots and autonomous vehicles to execute decisions with a higher degree of informed precision. The scope of 3D LiDAR applications is broad, encompassing autonomous vehicle navigation [39], forestry management [37], medical training simulations [39], smart city integrations [40], remote environmental sensing [41], and 3D-SLAM [42,43,44]. Conversely, 2D-LiDAR technology is predominantly employed for depth perception across various domains, such as in autonomous vehicular navigation and human tracking by robotic systems [45,46,47]. Notably, 2D-LiDAR sensors are both cost-effective and precise, offering the advantage of processing 2D point clouds without incurring significant computational overheads. However, a primary constraint of 2D-LiDAR is its limited environmental perception, restricted to a singular plane, which poses challenges in detecting individuals with a high degree of confidence. Despite this, it is imperative to acknowledge that 3D-LiDAR sensors, while offering expansive environmental insights, come with a higher price tag. This cost factor significantly impacts their feasibility for widespread deployment across multiple mobile robots or systems.

Recent developments have seen the integration of LiDAR with other modalities, such as 5G new radio (NR) positioning methods, providing a comprehensive approach to robot localization. This integration offers increased robustness in outdoor environments and serves as an additional absolute source of information when GNSS data are unreliable. The combination of LiDAR with 5G NR can lead to the development of more reliable and efficient SLAM frameworks, enhancing their capabilities in various domains such as healthcare, agriculture, city management, and public transportation [48]. There has been significant progress in developing efficient, consistent, and robust LiDAR-based SLAM algorithms. These methods, combined with deep-learning techniques for feature extraction and localization, have improved accuracy and robustness. Advances in optimization methods, such as nonlinear pose graph optimization, have further enhanced the consistency and efficiency of these algorithms. As a result, the adoption of LiDAR-based SLAM algorithms has increased in various real-world applications, particularly within the robotics and autonomous vehicle communities [49]. LiDAR presents a more reliable and efficient solution for positioning, mapping, and localization applications, particularly in challenging environments where radio signal-based methods such as GNSS and other RF technologies face limitations. The integration of LiDAR with emerging technologies such as 5G NR further underscores its potential for advancing robotic systems and autonomous navigation [50].

In this context, the aim of this work is to present the development of an ANN-based 2D-LiDAR pedestrian sensing and positioning system for B5G indoor applications. By developing a robust system that integrates the data collected from 2D-LiDAR sensors, this work aims to enhance the accuracy and reliability of indoor positioning and sensing. The subsequent sections describe the materials and methods, results, discussions, conclusions and future works.

## 2. Materials and Methods

### 2.1. ANN-Based 2D-LiDAR for Positioning B5G Applications

The deployment scenarios for an ANN-based 2D-LiDAR pedestrian sensing and positioning system are depicted in Figure 1. The sensor used is from the manufacturer SLAMTEC, from Shanghai, China. The model used is the RPLiDAR A1, which will be described in detail in the Section 2.2. LiDAR sensors are strategically positioned throughout the designated environment to facilitate pedestrian detection. The primary objective entails the transmission of data collected by the LiDAR sensors to a central processing unit, where it undergoes preliminary processing before being forwarded to the core network. Within the core network, an ANN algorithm is employed to accurately ascertain pedestrian locations. The identification of such locations enables the B5G core network to precisely control antenna array beamforming [51] and to configure RIS [52], thereby augmenting the user experience within the access network.

Specifically, Figure 1a illustrates a scenario within an indoor public space, wherein the data acquired from the LiDAR sensor system regarding the positions of individuals and objects are utilized by the B5G base station to conduct RIS and antenna array adjustments. Similarly, Figure 1b delineates an indoor factory environment. In both instances, the overarching aim is to significantly improve communication performance.

The architecture of the envisioned sensing and positioning system is illustrated in Figure 2. The process commences with the raw sensor data undergoing a conversion procedure, during which polar coordinates are transformed into Cartesian coordinates within a specified area of interest. These Cartesian coordinates, representing the detected points, are instrumental in identifying the presence of objects within each quadrant. Subsequently, these vectors serve as the input for the ANN. The output generated by the ANN is a vector, mirroring the dimensions of the input vectors, which delineates the detected pedestrian presence across various quadrants within the stipulated area of interest.

This granular positional information concerning individuals is subsequently disseminated to other integral components of the B5G core network. This dissemination facilitates the execution of critical operations such as RIS control, antenna array control, and the activation of emergency applications. Notably, such applications include coordination with rescue teams during disaster scenarios, thereby underscoring the system’s utility in enhancing safety and operational efficiency.

In indoor environments, conventional positioning techniques often exhibit significant imprecision, necessitating the integration of supplementary systems to enhance accuracy. In the context of advanced telecommunications, sensor data are transmitted to the B5G network core via a wireless access network. Within this framework, each sensor is treated as an individual network user, thereby consuming a portion of the network’s capacity allocated specifically for this application. The choice of 2D-LiDAR sensors is strategic, as they generate considerably less data, thus minimizing the impact on the network’s overall capacity. Additionally, the installation of the system requires the establishment of power supply points at strategic locations. At the network core, a dedicated computer is essential for processing the data in real time and disseminating this information to other applications, ensuring efficient and timely utilization of the positioning data.

### 2.2. The 2D-LiDAR

The RPLiDAR A1 presented in Figure 3 is a high-performance, compact-size time of flight (ToF) LiDAR using a laser centered at 785 nm, which is commercialized by SLAMTEC, from Shanghai, China. It features a 360° angular range and a measuring range from 0.15 to 12 m. The device provides a high sampling frequency of 12k samples/s and a rotational speed of 10 rpm. The dimensions of the RPLiDAR A1 are only 70 × 55 × 97 mm, making it suitable for integration into robotic systems.

The Slamtec RPLiDAR A1 represents a cost-effective and versatile LiDAR sensor that was primarily designed for indoor robotic applications, leveraging pulsed laser technology to measure distances and create detailed 2D representations of its surroundings. Characterized by its affordability and ease of integration, the RPLiDAR A1 supports a range of several meters for object detection, offers fine angular resolution for precise feature identification, and operates at high rotation speeds to enable real-time mapping and navigation. It could be applied in robotics for executing many tasks, such as navigation, room mapping, obstacle avoidance, and environment scanning.

### 2.3. LiDAR Data Processing

The data processing framework of the system is divided into two principal phases: data acquisition and data analysis. The initial phase, data acquisition, encompasses the deployment of LiDAR sensors to emit light pulses, thereby facilitating distance measurement. For this project, the sensor is strategically positioned indoors at an elevation of 1.3 m above ground level. The algorithms for sensor control and data logging have been meticulously developed in C++, leveraging a construct known as a named pipe. A named pipe functions as a transient software linkage between two entities (either programs or commands), acting akin to a virtual file. This virtual file temporarily houses data, enabling its transfer from one process to another in a unidirectional manner, thus serving as an efficacious medium for inter-process communication.

Proceeding to the data analysis phase, a Python 3.11.5 version script is employed to retrieve the data from the named pipes in real time for subsequent processing. This phase is dedicated to the processing and examination of the amassed LiDAR data to pinpoint object locations accurately. Measurements of distance and angle, initially in polar coordinates, are transformed into Cartesian coordinates. These transformed measurements are then systematically cataloged within a two-dimensional vector for further analysis.

### 2.4. Tests in Indoor Environments

A laboratory-based test setup was established to implement the proposed ANN-based positioning and sensing system. LiDAR sensors were strategically positioned in the 4 × 4 m square grid. Despite the sensor’s 12 m range capability, the focus was restricted to this grid. The grid was subdivided into sixteen 1 × 1 m squares, sequentially numbered from zero to 15, as shown in Figure 4. Scenario 1 is shown in Figure 4a, using one LiDAR sensor in the center of the area, and Scenario 2 is shown in Figure 4b, using two LiDAR sensors on opposite vertices.

Data acquisition from the LiDARs, presented in pairs of coordinates, passes through a preprocessing phase. This phase ensures the utilization of only those samples located within the predefined area since the LiDAR sensor is capable of taking measurements with a range of up to 12 m around it and the area of interest to be analyzed has dimensions smaller than its range region. The dataset is refined to exclusively include the data points that are within the predefined area of interest, as shown in Figure 5a for test scenario 1 and Figure 5b for test scenario 2, aiming to enhance the system efficiency.

### 2.5. ANN—Artificial Neural Network

#### 2.5.1. Input and Output Structures and Labels

The input for the neural network derives from LiDAR data, initially collected as point pairs of coordinates (x, y). The system uses the Cartesian coordinates of the detected points to determine the presence of objects in each quadrant, and these vectors are used as the ANN input parameters. Each input vector element corresponds to a specific quadrant of the mapped grid, with the value “1” indicating the presence of detected points in that quadrant and “0” in the case of not detecting points. The labels are the information provided along with the input data, representing the correct classification for each set of readings in terms of their location in the grid quadrants. Each element of the label vector corresponds to a specific quadrant within a mapped grid, with the value “1” indicating the presence of a pedestrian in that quadrant and “0” the absence. These labels are fundamental for supervised training because they provide the critical information that allows an algorithm to learn the relationship between input data and the desired output, serving as the reference against which the network adjusts its weights and biases to optimize predictions. In the same way, the output vector elements correspond to a specific quadrant within a mapped grid, with the value “1” indicating the presence of pedestrian’s readings in that quadrant and “0” the absence.

#### 2.5.2. Neural Network Architecture

The network structure presents layers with specific activation functions, reflecting a carefully balanced complexity. The input layer is composed of a number of neurons equal to the number of quadrants in the grid, 16; the other layers present the characteristics as follows:First Hidden Layer: This layer has 128 neurons, a choice that reflects the balance between computational capacity and sufficient complexity to capture a wide range of patterns in the data. The ReLU activation function is used here for its effectiveness in adding non-linearity to the model, allowing the network to learn complex and varied patterns in the input data.Second Hidden Layer: Contains 64 neurons, following the logic of a “funnel” architecture, where the progressive reduction in the number of neurons aids in consolidating learned patterns and reducing the complexity of the model, facilitating generalization to new data. The ReLU function continues to be used to maintain non-linearity.Output Layer: Composed of a number of neurons equal to the number of quadrants in the grid, 16, this layer uses the sigmoid activation function. This choice is ideal for binary classification, transforming the network outputs into probabilities that range from 0 to 1, interpreted as the probability of the presence of the reading in each specific quadrant.

Figure 6 presents the ANN layer diagram with the previously described structure.

The number of hidden layers and the number of neurons in each layer were defined through the problem conception, as well as prior experimentation and good design practices. For the input and output layers, the number of neurons is equal to the number of quadrants that the area is divided into. Additionally, the computational capacity of the system was taken into account to enable real-time application with the available equipment, thereby seeking a balance between computational cost and system performance.

For the training process, dropout layers were inserted after each hidden layer, with a rate of 0.2. These layers help to mitigate the risk of overfitting. They function by randomly deactivating a set of neurons during training, forcing the network to learn more robust and generalizable patterns.

#### 2.5.3. Training and Evaluation Process

Dividing the data into 80% for training and 20% for testing balances the need for a substantial sample for training with an adequate amount for validation and testing over 16,000 epochs. The model is compiled with the binary cross-entropy loss function and optimized with the Adam algorithm. This algorithm is an efficient optimizer for most neural network use cases. The performance of the model is evaluated using accuracy, precision, recall, and F1 score [54]. Table 1 presents the ANN layer specifications for the training process.

## 3. Results

The sensing and positioning experiments were conducted in a teaching laboratory from our institute (Inatel), as shown in Figure 7. An indoor scene with two people positioned within the area of interest and the poses identified by a person detection algorithm. The you only look once (YOLO) person detection algorithm is employed to accurately detect and determine the coordinates of the individuals within the environment, automatizing the labeling process. The experiments were conducted in two scenarios. In scenario 1, two pedestrians randomly altered their positions every 20 samples, aiming to emulate varied environmental conditions; the total number of samples was 1300; and just one LiDAR sensor was positioned in the center of the area. For scenario 2, four pedestrians randomly altered their positions every 20 samples; the total number of samples was 2000; and two LiDAR sensors were positioned on two opposite vertices of the area.

To mitigate sample redundancy and enhance the diversity of the dataset, specific measures were implemented during the experimental setup. In the first test scenario, two pedestrians were instructed to move in a completely random pattern, altering their positions every 20 samples. This approach also included the creation of various environmental conditions, such as partial or total shading, and different orientations of the pedestrians relative to the sensor, including facing forward, backward, sideways, and diagonally. A similar methodology was applied in the second test scenario, which involved a consistent presence of four pedestrians within the area of interest. Here, the position and orientation of each pedestrian were changed every 20 samples, again under conditions of partial or total shading. For both scenarios, data collection occurred at one-second intervals, ensuring a robust dataset for subsequent analysis. This strategy was designed to simulate realistic and challenging conditions for the positioning system, thereby providing a comprehensive assessment of its performance under varied environmental influences.

For both test scenarios, the samples were divided into 80% for training and 20% for validation testing. The samples were separated in the order in which they were generated. In test scenario 1, the first 1040 samples were used for training and the subsequent 260 samples for validation. In test scenario 2, the first 1600 samples were used for training and the next 400 samples for validation testing.

The system’s performance was assessed using two inputs: the label and input vectors. The system performance metrics were accuracy, precision, recall, and F1 score. Accuracy represents the overall correctness of the model, calculated by dividing the sum of true positives (TP) and true negatives (TN) by the total number of cases, which includes both correct and incorrect predictions (TP + TN + FP + FN) [28]. Precision focuses on the accuracy of positive predictions, determined by dividing TP by the sum of TP and false positives (FP), essentially measuring the quality of the positive class predictions [28]. Recall, on the other hand, assesses the model’s ability to detect positive instances by dividing TP by the sum of TP and false negatives (FN), highlighting the model’s sensitivity to detecting positive cases [28]. Lastly, the F1 score serves as the harmonic mean of precision and recall, offering a balance between them by calculating 2 times the product of precision and recall divided by the sum of precision and recall. This balanced approach is particularly useful when dealing with imbalanced classes, ensuring that both precision and recall are taken into account.

### 3.1. Test Scenario 1

The input vector was formulated using data obtained exclusively by the LiDAR sensor. The system output is a vector indicating which quadrants detected the presence of people. Table 2 presents the system performance evaluation in terms of accuracy, precision, recall, and F1 score. It is important to highlight that in this first scenario, only 1 LiDAR was used for sensing; it was positioned in the center of the area; 1300 samples were collected, with only 2 pedestrians moving randomly every 20 samples.

The outcomes delineated in Table 2 elucidate the system performance metrics, revealing an accuracy of 61.53%, which denotes a commendable level of model correctness. Notably, the precision metric stands at an impressive 90.38%, indicating that the system’s predictions regarding the presence of individuals are accurate in a significant majority of cases. Furthermore, the recall rate, quantified at 71.15%, signifies the system’s proficient capability in the identification of positive instances, thereby ensuring a robust capture of pertinent detections. The F1 score, mirroring the recall rate at 71.15%, embodies an optimal balance between precision and recall, a critical attribute in contexts where the omission of positive detections could entail substantial repercussions. The presented data accentuates the system’s pronounced strengths, particularly in terms of precision and recall, while concurrently acknowledging the potential for advancements in overall accuracy to amplify its utility and efficacy in real-world deployments.

### 3.2. Test Scenario 2

The input vector was formulated using data obtained from two LiDAR sensors at opposite vertices. The system output is a vector indicating which quadrants detected the presence of people. Table 3 presents the system performance evaluation in terms of accuracy, precision, recall, and F1 score, considering two cases: The data from the LiDAR sensors as independent input vectors for the ANN; and the data from the LiDAR sensors combined as just one input vector. It is important to highlight that in this case, 2 LiDAR sensors were used for sensing; these were positioned in separate areas of the area; 2000 samples were collected; with 4 pedestrians moving randomly every 20 samples.

The results, as detailed in Table 3, reveal that the combined data approach yielded superior performance, achieving an accuracy of 95.25%, precision of 98.787%, recall of 98.537%, and an F1 score of 98.571%. In contrast, the approach using no data combination reported slightly lower metrics, with an accuracy of 94.125%, precision of 98.329%, recall of 98.088%, and an F1 score of 98.137%. These findings underscore the enhanced effectiveness of integrating data from multiple LiDAR sensors for the purpose of human detection, as evidenced by the improved performance metrics across the board.

## 4. Discussion

This section presents a comparative analysis of the performance metrics obtained from two distinct experimental scenarios, focusing on the application of LiDAR sensors for human detection. The evaluation is structured around accuracy, precision, recall, and F1 score metrics, providing a comprehensive overview of the system’s effectiveness under varying conditions.

### 4.1. Scenario Comparison: Single vs. Dual LiDAR Configuration

The initial scenario employed a single LiDAR sensor positioned centrally within the detection area, yielding an accuracy of 61.53%. While the precision was notably high at 90.38%, the recall and F1 score were comparatively lower, at 71.15%. This configuration, despite its commendable precision, highlighted a significant gap in overall detection accuracy and recall, suggesting limitations in the sensor’s coverage and the system’s ability to generalize from the data. In contrast, the second scenario, which utilized two LiDAR sensors placed at opposite vertices without data combination, demonstrated a marked improvement across all metrics. Accuracy surged to 94.125%, accompanied by precision, recall, and F1 scores exceeding 98%. This enhancement can be attributed to the expanded coverage and increased data diversity, enabling the ANN to more effectively learn and predict the presence of individuals across the monitored area.

### 4.2. Impact of Data Combination in Dual LiDAR Configuration

Further analysis within the second scenario revealed that combining data from the two LiDAR sensors into a single input vector for the ANN resulted in even higher performance metrics: Accuracy reached 95.25%, with precision, recall, and F1 score all surpassing 98.5%. This improvement underscores the value of integrating sensing data to achieve a more comprehensive and nuanced representation of the environment, thereby enhancing the system’s predictive accuracy and reliability.

### 4.3. Comparative Analysis of Data Integration Strategies

The comparative evaluation between non-combined and combined data approaches within the dual LiDAR sensor scenario illustrates the nuanced benefits of data integration. While both strategies yielded high performance, the combined data approach slightly outperformed the non-combined configuration. This increment, albeit modest, is indicative of the potential for data fusion to mitigate detection blind spots and inconsistencies, offering a more holistic view of the detection space.

### 4.4. Final Comments

The transition from a single to a dual LiDAR sensor configuration, coupled with the strategic integration of sensory data, significantly enhances the system’s ability to detect human presence with high accuracy, precision, recall, and F1 scores. The findings from this study highlight the critical role of sensor placement and data processing techniques in optimizing detection systems for real-world applications. Future research should explore the scalability of these approaches to larger, more complex environments and the integration of additional sensory modalities to further improve detection capabilities.

## 5. Conclusions and Future Works

This work reported the development of an ANN-based 2D-LiDAR pedestrian sensing and positioning system for B5G applications. The proposed system aims to detect pedestrians and determine their location in indoor environments. Using their location data, such communication systems are capable of properly managing RIS and antenna arrays to direct the information signal towards these potential users in order to enhance their experience. Additionally, pedestrian locations in indoor environments can benefit from security and emergency systems in both smart buildings and cities.

The system development relied on the use of a 2D-LiDAR sensor, gathering a dataset of 1300 samples, split into 80% for training and 20% for performance testing. Experimental evaluation. Preprocessing, filtering, and combining these data in the core before feeding them into the ANN is crucial. For test scenario 1, the system performance highlights a moderate accuracy of 61.53% that suggests room for improvement. Moreover, the system demonstrates exceptional precision at 90.38%, indicating a high degree of reliability in its positive predictions. The recall rate of 71.15% further underscores the system’s effective capability in identifying relevant instances, ensuring that the majority of positive cases are detected. The F1 score, identical to the recall rate, reflects a harmonious balance between precision and recall, emphasizing the system’s adeptness at minimizing false positives while capturing true positives.

In test scenario 2, the system’s performance is evaluated considering two LiDAR sensors in the environment and conditions of no combination and combination data. The analysis reveals that employing a combined data strategy from multiple LiDAR sensors significantly enhances performance metrics in human detection tasks. Specifically, this integrated approach achieved an impressive accuracy of 95.25%, precision of 98.787%, recall of 98.537%, and an F1 score of 98.571%. In comparison, the methodology that did not utilize data amalgamation yielded marginally lower performance indicators, recording an accuracy of 94.125%, precision of 98.329%, recall of 98.088%, and an F1 score of 98.137%. These results emphatically highlight the superior efficacy of leveraging data from multiple LiDAR sources, as demonstrated by the uniformly improved performance metrics.

A critical consideration in this system is the preservation of pedestrian privacy, which is compromised by camera-based methods. Additionally, the system’s applicability in real-world scenarios represents a significant advancement. Experimental results indicate that integrating these sensors with ANNs improves the performance of indoor positioning systems. These findings also provide a foundation for future research in this area.

Future research directions for the project are outlined as follows: Firstly, expanding the dataset used for training and validation by incorporating a larger number of samples is anticipated to enhance the robustness of the model. Secondly, increasing the diversity of environmental conditions during data collection, specifically by varying the number of people present, is expected to improve the system’s adaptability and accuracy. Thirdly, the deployment of a greater number of spatially distributed 2D sensors within the area of interest will likely provide more comprehensive coverage and data fidelity. Additionally, the integration of recently acquired 3D LiDAR sensors is planned for subsequent phases of the project, aiming to refine depth perception and spatial analysis capabilities. Finally, modifications to the ANN are proposed, including alterations to the structure and activation functions of neurons in the hidden layers. A careful evaluation of the number of neurons per layer is also crucial, as it significantly influences the processing time of the system. These enhancements are intended to optimize the performance and efficiency of the positioning system in complex indoor environments.

## Figures and Tables

**Figure 1 micromachines-15-00620-f001:**
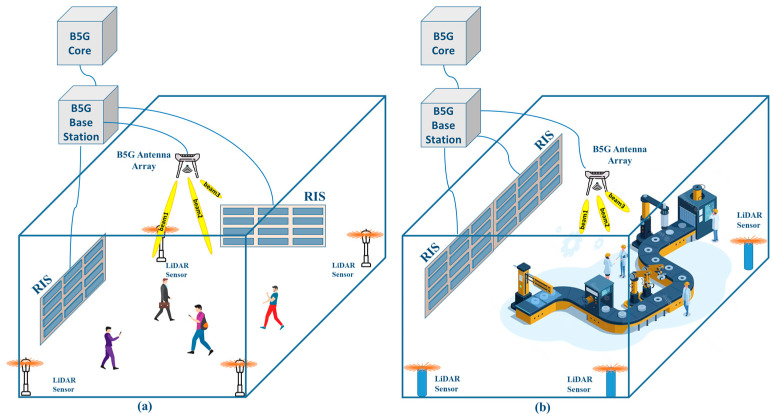
Application scenarios of the ANN-based 2D-LiDAR pedestrian sensing and positioning system: (**a**) indoor public environment; (**b**) indoor factory environment.

**Figure 2 micromachines-15-00620-f002:**
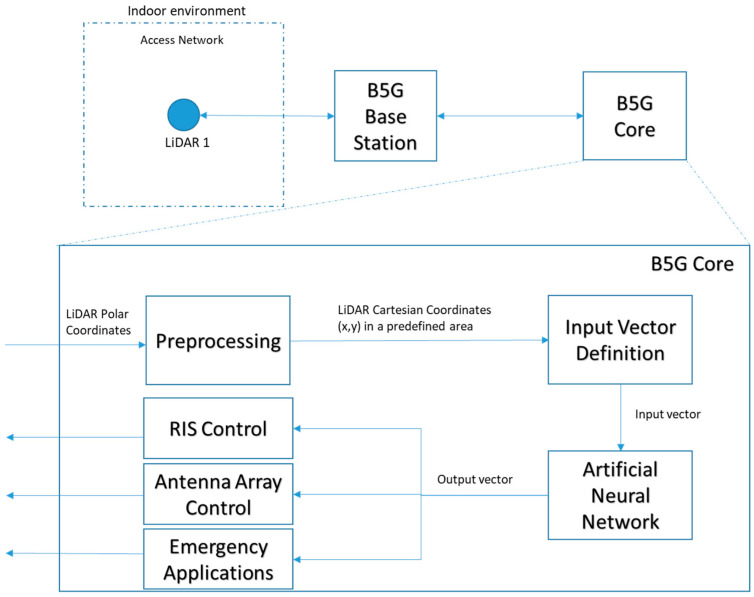
System architecture and block diagram.

**Figure 3 micromachines-15-00620-f003:**
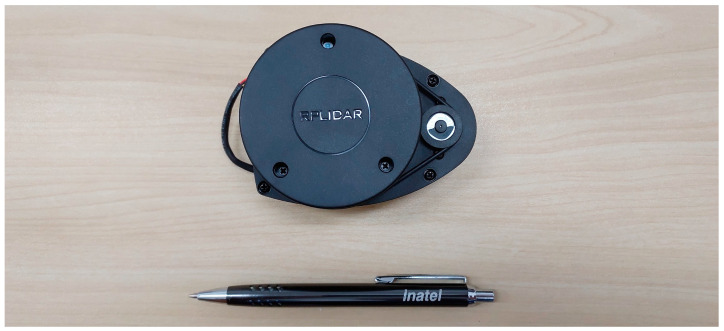
Slamtec LiDAR RPLiDAR A1 [53].

**Figure 4 micromachines-15-00620-f004:**
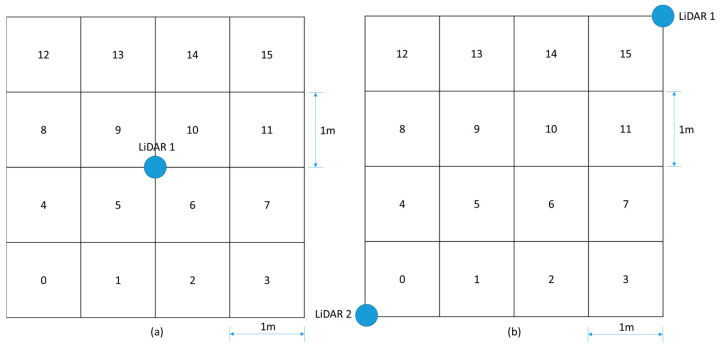
The tests environment setup and grids. (**a**) Test scenario 1. (**b**) Test scenario 2.

**Figure 5 micromachines-15-00620-f005:**
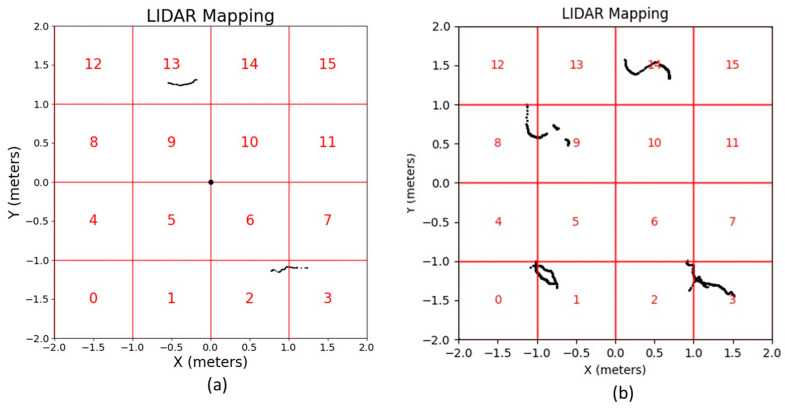
LiDARs data refined after preprocessing. (**a**) Test scenario 1. (**b**) Test scenario 2.

**Figure 6 micromachines-15-00620-f006:**
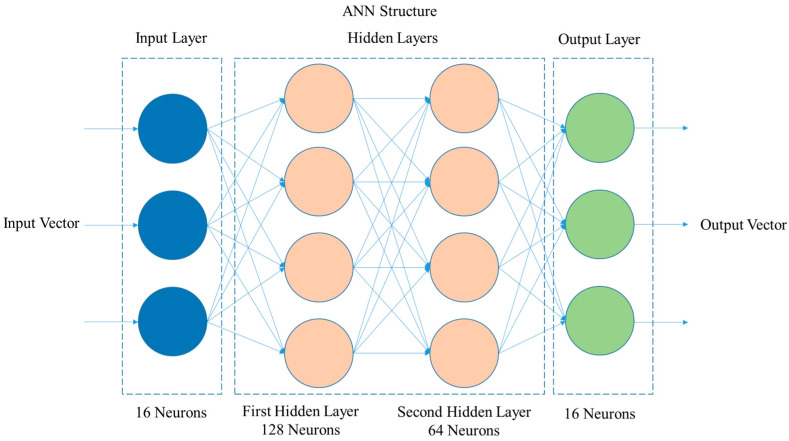
ANN layer diagram.

**Figure 7 micromachines-15-00620-f007:**
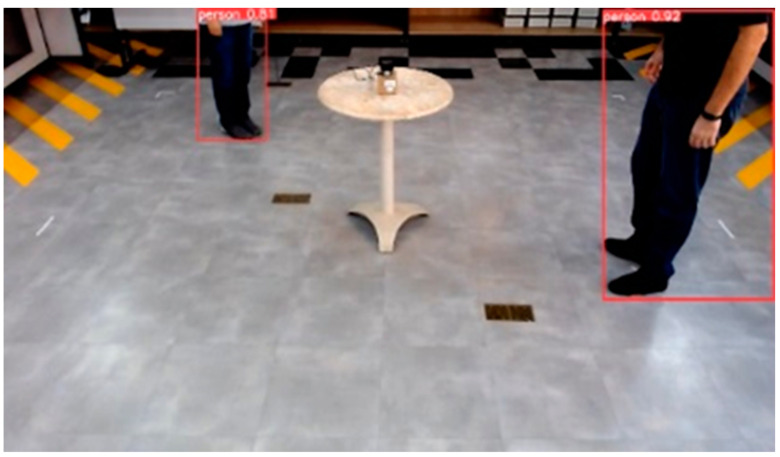
Teaching laboratory for evaluating and validating the proposed B5G sensing and positioning system using ANN-based 2D-LiDAR; their position on the ground for scenario 1 tests.

**Table 1 micromachines-15-00620-t001:** ANN layer specifications for the training process.

Layer	Neurons	Activation Function	Use	Observations
Input	16	-	Receives binary vectors of LiDAR readings and labels	Vectors represent presence/absence in quadrants
First hidden layer	128	ReLU	Captures more complex patterns in the data	-
Dropout (after the first hidden layer)	-	-	Prevents overfitting by forcing the network to learn more generalizable patterns	Rate of 0.2
Second hidden layer	64	ReLU	Consolidation of learned patterns	-
Dropout (after the second hidden layer)	-	-	Prevents overfitting by forcing the network to learn more generalizable patterns	Rate of 0.2
Output layer	16	Sigmoid	Produces classification probabilities per quadrant, returning 1 if there is a person in a quadrant	-

**Table 2 micromachines-15-00620-t002:** ANN performance evaluation in terms of accuracy, precision, recall, and F1 score for scenario 1.

Accuracy	Precision	Recall	F1 Score
61.53%	90.38%	71.15%	71.15%

**Table 3 micromachines-15-00620-t003:** ANN performance evaluation in terms of accuracy, precision, recall, and F1 score for scenario 2.

Data	Accuracy	Precision	Recall	F1 Score
No Combination	94.125%	98.329%	98.088%	98.137%
Combination	95.25%	98.787%	98.537%	98.571%

## Data Availability

The original data presented in the study are openly available in repository at https://drive.google.com/drive/folders/1Ct-xbJOJk2T5s7286zpcb4Vo_no68Cmf?usp=drive_link.

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
