# Peer review of "ANN-Based LiDAR Positioning System for B5G"

_micromachines, 2024, doi:10.3390/mi15050620_

Round 1

Reviewer 1 Report

Comments and Suggestions for Authors

It is an interesting and novel approach to indoor positioning that leverages 2D LiDAR (Light Detection and Ranging) technology combined with an Artificial Neural Network (ANN). The authors have laid a strong foundation, discussing the current limitations of Indoor Positioning Systems (IPS) and emphasizing the necessity of integrating advanced sensing capabilities into the future generation of mobile networks for enhanced user experience. They propose a system that utilizes 2D-LiDAR sensors in conjunction with digital filters, a camera, and an image-processing algorithm to train the ANN, aiming to optimize B5G network operations.

I recommend publication subject to some revisions, as detailed later in this review.

Here are some comments/queries.

  1. The system demonstrates high precision but more moderate accuracy and recall. What are the factors contributing to this discrepancy, and how could the model be improved to increase its overall accuracy?
  2. Can you provide more details on the testing conditions, such as how representative the sample of 1300 data points is for real-world scenarios, and how the model’s performance might scale with a larger dataset?
  3. How will the system’s reliance on precise positioning impact the deployment and scalability within B5G networks, and what infrastructure is required to support this system?
  4. What specific future works are planned to address the noted limitations of the system, such as the use of 3D-LiDAR sensors or different ANN structures?

Reviewer 2 Report

Comments and Suggestions for Authors

In this study, the authors report the development of an efficient and precise indoor positioning system utilizing two-dimensional (2D) Light Detection and Ranging (LiDAR) technology, aiming to address the challenging sensing and positioning requirements of the Beyond Fifth Generation (B5G) mobile networks. However, there are still some doubts that need to be further explained.

1、I don't understand the purpose of Figure 5. What is the difference between what it wants to express and Figure 4? Don’t you just want to express the sensing range of the LiDAR sensor?

2、In the Neural Network Architecture section, it is recommended that the author add a network structure diagram to show the model structure more clearly.

3、The author mentioned that 1300 samples were obtained, but are they always composed of two pedestrians? Will the richness of the samples be insufficient, causing the sample similarity to be too high, resulting in a higher accuracy rate?

4、Based on the samples obtained above, how are the training samples and test samples divided?

5、There are references that have not been correctly cited.

Comments on the Quality of English Language

Moderate editing of the English language required

Round 2

Reviewer 2 Report

Comments and Suggestions for Authors

There are no further questions.

Comments on the Quality of English Language

Minor editing of English language required